# Se-Enriched *Cardamine violifolia* Improves Laying Performance and Regulates Ovarian Antioxidative Function in Aging Laying Hens

**DOI:** 10.3390/antiox12020450

**Published:** 2023-02-10

**Authors:** Hui Wang, Xin Cong, Kun Qin, Mengke Yan, Xianfeng Xu, Mingkang Liu, Xiao Xu, Yue Zhang, Qingyu Gao, Shuiyuan Cheng, Jiangchao Zhao, Huiling Zhu, Yulan Liu

**Affiliations:** 1Hubei Key Laboratory of Animal Nutrition and Feed Science, School of Animal Science and Nutritional Engineering, Wuhan Polytechnic University, Wuhan 430023, China; 2Enshi Se-Run Material Engineering Technology Co., Ltd., Enshi 445000, China; 3National R&D Center for Se-Rich Agricultural Products Processing, School of Modern Industry for Selenium Science and Engineering, Wuhan Polytechnic University, Wuhan 430023, China; 4Department of Animal Science, Division of Agriculture, University of Arkansas, Fayetteville, NC 72701, USA

**Keywords:** *Cardamine violifolia*, hens, selenium, ovary, laying performance, antioxidant

## Abstract

As a selenium-enriched plant, *Cardamine violifolia* (SEC) has an excellent antioxidant function. The edibility of SEC is expected to develop new sources of organic Se supplementation for human and animal nutrition. This study was conducted to investigate the effects of SEC on laying performance and ovarian antioxidant capacity in aging laying hens. A total of 450 laying hens were assigned to five treatments. Dietary treatments included the following: a basal diet (diet without Se supplementation, CON) and basal diets supplemented with 0.3 mg/kg Se from sodium selenite (SS), 0.3 mg/kg Se from Se-enriched yeast (SEY), 0.3 mg/kg Se from SEC, or 0.3 mg/kg Se from SEC and 0.3 mg/kg Se from SEY (SEC + SEY). Results showed that supplementation with SEC tended to increase the laying rate, increased the Haugh unit of eggs, and reduced the FCR. SEC promoted ovarian cell proliferation, inhibited apoptosis, and ameliorated the maintenance of follicles. SEC, SEY, or SEC + SEY increased ovarian T-AOC and decreased MDA levels. SEC increased the mRNA abundance of ovarian *selenoproteins*. SEC and SEC + SEY increased the mRNA abundance of *Nrf2*, *HO-1*, and *NQO1*, and decreased the mRNA abundance of *Keap1*. These results indicate that SEC could potentially to improve laying performance and egg quality via the enhancement of ovarian antioxidant capacity. SEC exerts an antioxidant function through the modulation of the Nrf2/Keap1 signaling pathway.

## 1. Introduction

Aging is a normal, dynamic, and irreversible biological process, which can lead to a decline in organ, tissue, and cell function [1]. However, ovarian aging occurs both earlier and more rapidly than in other tissue [2]. As in mammals, laying hens are more prone to ovarian senescence due to increasing laying frequency. Ovarian aging is characterized by follicular atresia and decreases in both the quantity and the quality of oocytes [3]. In the poultry industry, ovarian senescence may be the main reason for the reduction in egg production and egg quality [4]. Moreover, the decline in egg production has brought about a huge loss of income to the poultry industry. Therefore, it is necessary to explore effective measures to alleviate ovarian recession.

However, the mechanism of ovarian aging is still not fully understood. Numerous studies have demonstrated that oxidative stress, induced by the accumulation of reactive oxygen species (ROS), is one of the most dominant factors [5,6]. The dysfunction and apoptosis of granulosa cells and age-related decline in female fertility are associated with oxidative stress. The nuclear factor erythroid 2-related factor 2 (Nrf2) and the Kelch-like ECH-associated protein 1 (Keap1) systems are important defense mechanisms against oxidative stress in vivo and in vitro [2,7]. Normally, Nrf2 binds to Keap1 in the cytoplasm and exists as an inactive form. When oxidative stress is triggered, Nrf2 is released from Keap1 and then transferred into the nucleus, ultimately activating the expression of antioxidative enzymes such as glutathione peroxidase (GSH-PX) and superoxide dismutase (SOD).

Based on this, alleviating ovarian aging by reducing oxidative stress has been researched in poultry. Antioxidant compounds such as vitamins and plant extracts have been applied to reduce oxidative stress in the ovaries and attenuate follicular atresia [2,8,9]. As an essential trace element with a strong antioxidant effect, selenium (Se) protects the organism against the actions of free radicals [10,11]. Some studies have suggested that a diet supplemented with different Se sources could promote the antioxidant capacity of laying hens and reduce the apoptosis in the ovary [12,13].

*Cardamine violifolia* is a Se-tolerant plant found in Enshi, Hubei, China. It has been shown to have Se content exceeding 700 mg/kg (dry weight) in the leaves, with over 85% of the complete Se deposited in the form of natural Se. Emerging studies have shown that Se-enriched *Cardamine violifolia* (SEC) prevents obesity and metabolic disorders induced by a high-fat diet in mice through ameliorating oxidative stress and inflammation [14]. Our laboratory also found that SEC supplementation improved growth performance and antioxidant capacity in broilers and weaned pigs [15,16]. However, there are no studies related to the effects of SEC on reproductive function in aging animal models. Therefore, in this study, we used an aging hen model. Our objective was to explore whether SEC treatment would improve laying performance via the enhancement of ovarian antioxidant capacity, acting to retard ovarian aging. Moreover, the role of the Nrf2/Keap1 pathway was investigated to clarify the possible mechanism of SEC-regulated ovarian aging.

## 2. Materials and Methods

### 2.1. Animal and Experimental Design

All animal experimental protocols (WPU202204006) were approved by the Animal Care and Use Committee of Wuhan Polytechnic University. A total of four hundred and fifty (65-week-old) Roman laying hens with similar reproductive performance were assigned to 1 of 5 dietary treatments. Each treatment contained 6 replicates of 15 birds. Dietary treatments included the basal diet (low-Se diet without Se supplementation, CON) and basal diets supplemented with 0.3 mg/kg Se from sodium selenite (SS), 0.3 mg/kg Se from Se-enriched yeast (SEY), 0.3 mg/kg Se from SEC, or 0.3 mg/kg Se from SEC and 0.3 mg/kg Se from SEY (SEC + SEY). The basal diet was formulated to meet or exceed the requirements of laying hens recommended by the National Research Council (without adding exogenous Se) [17]. The composition and nutrient levels of the corn–soymeal basal diet are shown in Table 1. The experimental period was 8 weeks.

The Se-enriched *Cardamine violifolia* (1430 mg/kg total Se content) used in this study was obtained from Enshi Se-Run Material Engineering Technology Co., Ltd., Enshi, China. The SS and SEY were purchased from Angel Yeast Co., Ltd. (Yichang, China).

### 2.2. Laying Performance

Egg production and egg weight were recorded daily to calculate the laying rate and the egg mass production (g/d/hen). Feed intake was recorded according to replicates every 2 weeks. The feed conversion ratio (FCR) was calculated as the ratio of feed consumed (g)/egg weight (g).

### 2.3. Egg Quality

Twenty-four freshly laid eggs were randomly collected from hens in each treatment (4 eggs/replicate) for the determination of egg quality at d 28 and 56 of the experimental period. Egg length, egg width, yolk width, and yolk height were measured. The egg shape index was calculated as (egg length/width) × 100. Eggshell strength was evaluated using the Egg Force Reader (Orka Technology Ltd., Ramat Hasharon, Israel). Egg weight, albumen height, Haugh unit, and yolk color were measured using the Egg Analyzer (Orka Technology Ltd., Ramat Hasharon, Israel). The yolk was separated and weighed. The yolk percentage was calculated as g yolk/g egg.

### 2.4. Sample Collection

At the end of the experiment, 6 hens per treatment (one hen from each replicate) were weighed and slaughtered. The ovaries were collected and weighed. The ovarian indices (ovary weight (g)/body weight (g) × 100%) were calculated. The numbers of preovulatory follicles (POFs, >10 mm), small yellow follicles (SYFs, 8 to 10 mm), and large white follicles (LWFs, 6 to 8 mm) were measured. Ovarian tissues without follicles of over 1 mm were separated into three parts. Two parts of the ovary were frozen in liquid nitrogen immediately and then stored at −80 °C for analyzing the redox state and mRNA expression. The other parts of the ovarian samples were fixed in 4% paraformaldehyde for histopathological examination and immunological staining.

### 2.5. Histology

The fixed ovaries were embedded in paraffin and sliced into 4 μm sections. The slides were stained with hematoxylin and eosin (H&E) using standard protocols. Stained slides were evaluated by two independent blinded researchers under light microscopy. The number of atretic follicles and normal follicles in each slide was counted, respectively. The histological criteria for normal follicles and follicular atresia were implemented in compliance with protocols described previously [18]. The follicular atretic rate was calculated as follows: follicular atretic rate (%) = number of atretic follicles/number of total follicles × 100.

### 2.6. Lipid Peroxidation and Antioxidant Enzyme Activity

The activity of GSH-Px and total SOD (T-SOD), the total antioxidant capacity (T-AOC), and the concentration of malonaldehyde (MDA) in the ovary were determined using commercially available kits (Nanjing Jiancheng Bioengineering Institute, Nanjing, China) according to the manufacturer’s protocols.

### 2.7. Real-Time Quantitative PCR

Total RNA was extracted from ovaries using Trizol reagent (Invitrogen) according to the manufacturer’s instructions. The complementary DNA was synthesized using the PrimeScript RT reagent kit with gDNA eraser (TaKaRa, #RR047A). The quantitative real-time PCR was performed using the SYBR Premix Ex Taq (Tli RNase H Plus) qPCR kit (TaKaRa, #RR420A) and 7500 Real-Time PCR system (Applied Biosystems). The expression of the target genes relative to the housekeeping gene (β-actin) was analyzed by the 2^−ΔΔCT^ method of Livak and Schmittgen [19]. Relative mRNA abundance of each target gene was normalized to the control group. Primer sequences are given in Table 2.

### 2.8. Immunofluorescence Staining

Immunofluorescence staining was carried out as previously reported [20]. Ovarian tissue sections were deparaffinized in xylene and rehydrated in reducing concentrations of ethanol. Antigen retrieval was conducted in a 10 mM sodium citrate buffer for 20 min at 100 °C. Tissue sections were incubated with rabbit anti-PCNA polyclonal antibody (1:200, Proteintech, Chicago, IL, USA) overnight at 4 °C after blocking with 5% goat serum. Then, slides were incubated with goat anti-rabbit secondary antibody (1:500) conjugated to CY3 (Invitrogen, Carlsbad, CA, USA) for 1 h at 37 °C. Subsequently, the nuclei were stained with 4′, 6-diamisino-2-phenylindole (DAPI, Boster Bioengineering Co., Ltd., Wuhan, China) and the slides were imaged on a laser confocal microscope (Olympus, Tokyo, Japan). The number of PCNA-positive cells (red) was counted and expressed as a percentage of the PCNA-labeled cells over the total number of ovarian cells in the same field (PCNA index).

### 2.9. TUNEL Assay

The TdT-mediated dUTP nick-end labeling (TUNEL) assay was conducted using a TUNEL assay kit (Vazyme, Nanjing, China) according to the manufacturer’s protocols. The number of TUNEL-positive cells (green) was counted and expressed as a percentage of the green-labeled cells over the total number of ovarian cells (TUNEL index).

### 2.10. Statistical Analysis

The data were analyzed using SPSS 16.0 statistical software (SPSS Inc., Chicago, IL, USA). The differences between dietary treatments were evaluated using one-way analysis of variance (ANOVA) followed by Duncan’s multiple comparison test. In addition, data for laying performance and egg quality were analyzed using repeated measure analysis in the general linear model to evaluate the effects of time and different treatments. *p* < 0.05 was defined as significant, while *p* < 0.10 was considered to be a trend toward significance.

## 3. Results

### 3.1. Laying Performance

The laying performance of hens is presented in Table 3. The laying rate tended to increase (*p* < 0.1) in hens fed the SEC-supplemented diet compared with the CON group. Moreover, hens fed the SEC diet had a tendency to increase (*p* < 0.1) the laying rate compared with those fed the SEY and SEC + SEY diet. Dietary supplementation with SEC decreased FCR (*p* < 0.05) compared with hens fed the CON diet during d 1–28 and d 1–56. Hens fed the SS and SEC + SEY diet had lower FCR (*p* < 0.05) compared with those fed the CON diet during d 1–56. Dietary treatment did not influence (*p* > 0.05) other indices of laying performance. In addition, average daily egg mass and ADFI decreased (*p* < 0.05) and average egg weight increased (*p* < 0.05) from d 1–28 to d 28–56.

### 3.2. Egg Quality

Dietary SEC, SEC + SEY, and SS supplementation increased (*p* < 0.05) the Haugh unit compared with hens fed the CON diet at d 28 (Table 4). Dietary supplementation with SEC and SS increased (*p* < 0.05) eggshell strength compared with hens fed the SEY diet at d 56. Furthermore, the egg yolk index and yolk percentage increased (*p* < 0.05) from d 28 to d 56. No differences were observed in other egg quality indices among dietary treatments (*p* > 0.05).

### 3.3. Ovarian Follicle Development

The follicle number is shown in Table 5. Dietary SEC supplementation increased (*p* < 0.05) the number of LWFs compared with hens fed the CON diet. There was no difference in the number of POFs, SYFs, and ovary index among dietary treatments (*p* > 0.05). HE staining showed that the number of primary and developing follicles in the CON group decreased (Figure 1). However, the follicular atretic rate in the CON group was higher (*p* < 0.05) than that in the SEC + SEY group (Figure 1). There was no difference in follicular atretic rate among all Se treatments (*p* > 0.05).

### 3.4. Proliferation and Apoptosis in Ovary

PCNA fluorescence staining was used to assess cell proliferation (PCNA-positive cells), while the TUNEL assay revealed cell apoptosis (TUNEL-positive cells). Hens fed the Se-supplemented diet had a higher (*p* < 0.05) relative PCNA index in ovarian tissue compared with those fed the CON diet (Figure 2A,B). The relative PCNA index in the SEC + SEY group was higher (*p* < 0.05) than that in the SS group. There was no difference in the relative PCNA index among the SEY, SEC, and SEC + SEY groups (*p* > 0.05). The TUNEL assay showed that dietary supplementation with SEY, SEC, and SEC + SEY decreased the relative TUNEL index of the ovary compared with hens fed the CON diet (Figure 2C,D). Similarly, dietary supplementation with SEC + SEY decreased (*p* < 0.05) the mRNA abundance of ovarian BCL2-associated X (*Bax*) compared with hens fed the CON, SS, and SEY diets (Figure 2E). Dietary SEC supplementation decreased (*p* < 0.05) ovarian *Caspase 3* mRNA abundance in comparison with those fed the CON and SS diets (Figure 2F). Dietary Se supplementation increased (*p* < 0.05) the mRNA expression of B-cell lymphoma-2 (*Bcl-2*) compared with those fed the CON diet (Figure 2G). Moreover, SEC supplementation increased (*p* < 0.05) the mRNA expression of *Bcl-2* compared with those fed the SEY or SEC + SEY diet.

### 3.5. Antioxidant Status

Dietary Se supplementation increased (*p* < 0.05) T-AOC in the ovary compared with hens fed the CON diet (Table 6). SEC + SEY supplementation had a tendency to increase (*p* < 0.1) the ovarian GSH-PX activity compared with the CON diet. Dietary SEY and SEC supplementation decreased (*p* < 0.05) the MDA content of the ovary in comparison with hens fed the CON diet. In addition, dietary treatment had no effect (*p* > 0.05) on ovarian T-SOD activity.

### 3.6. Relative mRNA Expression of Selenoproteins and Nrf2/Keap1 Pathway-Related Genes in the Ovaries of Hens

Dietary Se supplementation increased (*p* < 0.05) the mRNA abundance of ovarian selenoprotein f (*Selenof*) and glutathione peroxidase 1 (*GPX1*) compared with hens fed the CON diet (Figure 3A,B). Furthermore, hens fed the SEC diet had higher (*p* < 0.05) mRNA expression of ovarian *Selenof* than those fed the SS, SEY, and SEC + SEY diets. There was no difference in the mRNA expression of ovarian *GPX1* among all Se treatments (*p* > 0.05). Dietary SEC supplementation increased (*p* < 0.05) the mRNA abundance of ovarian glutathione peroxidase 1 (*GPX4*) compared with those fed the CON, SS, SEY, and SEC + SEY diets (Figure 3C).

Dietary SEC and SEC + SEY supplementation increased (*p* < 0.05) the mRNA abundance of ovarian *Nrf2* in comparison with hens fed the CON diet (Figure 3D). No differences were observed in *Nrf2* mRNA expression among the SEC, SEC + SEY, and SS groups (*p* > 0.05). In contrast, hens fed the SEC diet had lower (*p* < 0.05) mRNA abundance of ovarian *Keap1* compared with those fed the CON diet (Figure 3E). There was no difference in *Keap1* mRNA expression among all Se treatments (*p* > 0.05). Dietary SEY and SEC supplementation increased (*p* < 0.05) the mRNA abundance of ovarian NAD(P)H quinone dehydrogenase 1 (*NQO1*) compared with hens fed the CON diet (Figure 3F). In addition, hens fed the SEY, SEC, and SEC + SEY diets had higher (*p* < 0.05) heme oxygenase 1 (*HO-1*) mRNA expression compared with hens fed the CON diet (Figure 3G). There was no difference in *HO-1* mRNA expression among the SEY, SEC, and SEC + SEY diets (*p* > 0.05).

## 4. Discussion

Aging is related to the structural and functional alterations of all human organs [21]. Ovarian aging is accompanied by a decrease in ovarian follicle reserves and a decline in oocyte quality [21,22]. In poultry, ovarian recession shortens the lifespan of ovarian function and reduces the commercial value of laying hens [2]. Numerous studies have demonstrated that one of the main driving factors of ovarian aging is oxidative stress [13,23,24]. Therefore, alleviating oxidative stress in the ovaries might be an important breakthrough for retarding ovarian aging.

It was well known that Se plays a critical role in the tissue antioxidation system. It has been reported that diet Se supplementation is essential to maintain the performance of laying hens [25]. Generally, there are two major sources of Se additives for poultry, namely inorganic Se (mainly sodium selenite) and organic Se (mainly Se–yeast) [26]. Many studies have established that organic Se improves antioxidant properties and bioavailability [12,27]. SEC contains organic Se, which mainly exists in the form of Se-enriched protein. The edibleness and rapid accumulation of Se from SEC is expected to support the further development of new organic Se sources as supplementation for human and animal nutrition. In the present study, SS and SEY supplementation did not affect laying performance and egg quality. This is consistent with previous reports that the laying performance and egg quality of hens fed with different sources of Se was not different [12,28,29,30]. However, dietary SEC supplementation tended to increase the laying rate and decreased the FCR of laying hens. Similar results have not been reported so far. Furthermore, SEC or SEC + SEY supplementation increased the Haugh unit. These results suggested that SEC could potentially improve laying performance and egg quality in the late phase.

Ovarian recession of aging laying hens is regarded as one of the highest risk factors leading to a decline in egg production and egg quality [31]. The laying performance of hens depends on the number of ovarian follicles. Liu et al. reported [32] that the number of follicles in laying hens decreased sharply from d 280 to 580, while the number of atretic follicles increased. Our results showed that the follicular atretic rate in the CON group increased compared to that in the SEC + SEY group. Moreover, dietary SEC supplementation increased the number of LWFs, which is consistent with the change in the laying rate. Follicular atresia is an apoptotic process that is modulated by proapoptotic factors and antiapoptotic factors [4]. As a proapoptotic factor, *Bax* participates in initiating the apoptosis program. The active apoptosis signal leads to a series of downstream caspase cascades and induces the apoptosis of granulosa cells in the early stage of follicular atresia [33]. In contrary, *Bcl-2* binding to *Bax* can prevent apoptosis [34]. In the present study, dietary supplementation with different Se sources increased the mRNA expression of ovarian *Bcl-2.* Moreover, SEC or SEC + SEY supplementation decreased the mRNA abundance of ovarian *Bax* and *Caspase 3.* Meanwhile, we also found that SEC or SEC + SEY supplementation improved proliferation rate and decreased apoptosis rate in ovarian cells. These results indicated that SEC alleviated ovarian aging and could potentially improve the laying rate by modulating the proliferation and apoptosis of ovarian cells.

ROS gradually increases with aging, which contributes to poor fertility in aged females [35,36]. Oxidative stress occurs when ROS production exceeds the antioxidant defense capacity of cells [37]. The endogenous antioxidants include SOD, GSH-PX, and T-AOC, which play an important role in protecting the cellular structures from the damage of ROS induced by aging in laying hens [38]. MDA is a metabolic product of lipid peroxidation and is a biomarker of oxidative stress. GSH-PX is a Se-dependent enzyme that catalyzes the reduction of hydrogen peroxide and organic peroxides to water [39]. In the present study, dietary Se supplementation increased T-AOC in the ovary. This is consistent with the results of previous studies [30,40]. Our study showed that SEC, SEY, and SEC + SEY tended to elevate GSH-PX activity and reduced MDA levels in the ovaries of hens. In agreement with our results, Jing et al. reported [41] that hens fed organic Se had increased GSH-PX activity and decreased MDA content in plasma. This may be attributed to the fact that the organic sources of Se (SEC and SEY) mainly exist in the form of selenoproteins and have high bioavailability. Se exerts its antioxidant function by incorporation into selenoproteins. It is believed that selenoproteins such as GPX1, GPX3, GPX4, and Selenof can eliminate the accumulating ROS and mitigate the oxidative stress during ovarian follicle development [42]. In our study, dietary supplementation with different Se sources up-regulated the mRNA expression of *Selenof* and *GPX1* in the ovary. SEC supplementation also increased the mRNA expression of ovarian *GPX4*. In agreement with our study, Yang et al. indicated [13] that dietary Se elevated the mRNA expression of *GPX1*, *GPX3*, *GPX4*, and *Selenof* in ovaries of aging mice, ameliorating the ovarian oxidative stress induced by aging. Thus, our results indicated that SEC improved the ovarian antioxidant capacity, alleviating ovarian aging.

The Nrf2/Keap1 signaling pathway plays an important role in the resistance to oxidative stress [7,43]. As a key factor, Nrf2 activates the oxidative stress defense system to regulate the transcription of antioxidant genes, such as *SOD*, *GSH-PX*, and catalase (*CAT*) [44]. Keap1 is a negative regulator of Nrf2 [45]. Liu et al. indicated [32] that the Nrf2/Keap1 pathway was down-regulated during the ovarian aging process. Reszka et al. demonstrated [46] that plasma Se levels were negatively correlated with *Keap1* mRNA levels and positively correlated with *Nrf2* mRNA levels in the peripheral blood leukocytes of humans. Similar to previous research, our study indicated that SEC or SEC + SEY supplementation up-regulated the mRNA expression of *Nrf2* and down-regulated that of *Keap1* in the ovaries of laying hens. Furthermore, SEC and SEY supplementation increased Nrf2/Keap1 downstream antioxidant enzyme signals such as *HO-1* and *NQO1* mRNA expression in the ovary. These results suggested that the antioxidant effect of SEC might be associated with the activation of the Nfr2/Keap1 signaling pathway.

## 5. Conclusions

In conclusion, dietary Se-enriched *Cardamine violifolia* enhanced ovarian antioxidant capacity through the modulation of the Nrf2/Keap1 signaling pathway. This suggested that dietary supplementation with *Cardamine violifolia* might alleviate ovarian senescence, and potentially improve the laying rate and egg quality in aging laying hens. Furthermore, this study provides evidence for the development and application of *Cardamine violifolia* as a feed additive to improve the laying performance and egg quality in laying hens.

## Figures and Tables

**Figure 1 antioxidants-12-00450-f001:**
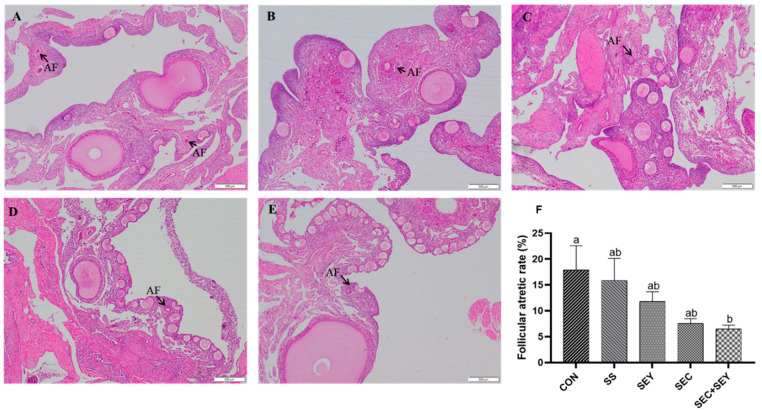
Effects of different Se sources on the histology of ovary in laying hens. (**A**–**E**) H&E staining of the ovaries. (**A**) the CON group; (**B**) the SS group; (**C**) the SEY group, (**D**) the SEC group, and (**E**) the SEC + SEY group. Scale bar = 200 μm. AF, atresia follicle. (**F**) Effects of different Se sources on follicular atretic rate. Data are means of 6 replicates per dietary treatment. Bars with different letters (a,b) indicate a significant difference (*p* < 0.05).

**Figure 2 antioxidants-12-00450-f002:**
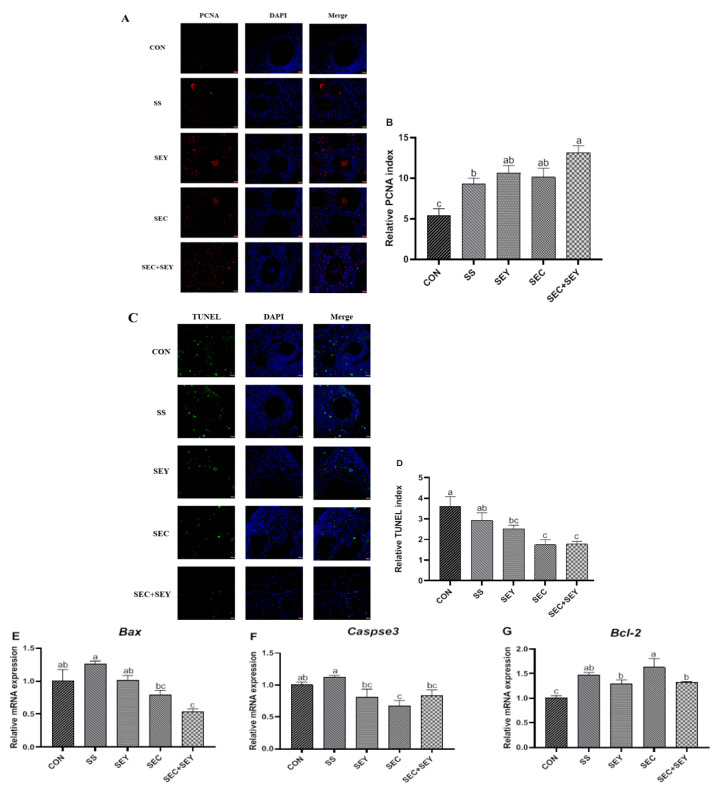
Effects of different Se sources on cell proliferation and apoptosis of ovaries in laying hens. (**A**): PCNA staining of the ovaries. (**B**): Quantification of PCNA-positive cells. (**C**): TUNEL staining of the ovaries. (**D**): Quantification of TUNEL-positive cells. All cell nuclei showed blue fluorescence, indicating DAPI staining. The PCNA-labeled cells exhibited red fluorescence, showing the proliferative cells. The TUNEL-labeled cells exhibited green fluorescence, showing apoptosis. Scale bar = 10 µm. (**E**–**G**): Effects of different Se sources on the expression of apoptosis-related genes in the ovary of laying hens. Data are means of 6 replicates per dietary treatment. Bars with different letters (a–c) indicate a significant difference (*p* < 0.05). Bax, BCL2-associated X; Bcl-2, B-cell lymphoma-2.

**Figure 3 antioxidants-12-00450-f003:**
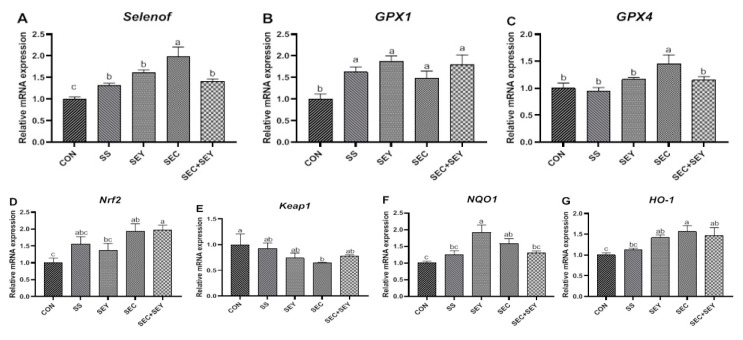
Relative mRNA expression of selenoprotein (**A**–**C**) and Nrf2/Keap1 pathway genes (**D**–**G**) in the ovaries of laying hens fed different Se sources. Data are means of 6 replicates per dietary treatment. Bars with different letters (a–c) indicate a significant difference (*p* < 0.05). Selenof, selenoprotein f; GPX1, glutathione peroxidase 1; GPX4, glutathione peroxidase 4; Nrf2, nuclear factor erythroid 2-related factor 2; Keap1, Kelch-like ECH-associated protein 1; NQO1, NAD(P)H quinone dehydrogenase 1; HO-1, heme oxygenase 1.

**Table 1 antioxidants-12-00450-t001:** Composition and nutrient levels of basal diet for laying hens (air dry basis).

Ingredients			Nutrient Levels ^2^	
Corn	63.25		Metabolizable energy (MJ/kg)	11.62
Soybean meal	25.45		Crude protein	16.31
Limestone	8.00		Calcium	3.66
Calcium hydrogen phosphate	1.50		Available phosphorus	0.41
Sodium chloride	0.30		Lysine	0.35
Premix ^1^	0.50		Selenium ^3^ (mg/kg)	0.156
DL-methionine	0.10			

^1^ The premix provided the following (per kilogram of diet): vitamin A, 8000 IU; vitamin D3, 3000 IU; vitamin E, 20 IU; vitamin K, 2.5 mg; cobalamine, 23 μg, pantothenate, 8 mg; folacin, 1 mg; vitamin B1, 2.5 mg; vitamin B2, 5.5 mg; niacin, 30 mg; vitamin B6, 4 mg; vitamin B12, 20 mg; biotin, 55 μg; choline chloride, 500 mg; iron, 90 mg; copper, 8 mg; zinc, 80 mg; manganese, 90 mg; iodine, 0.6 mg. ^2^ Nutrient levels are calculated values. ^3^ Se level is measured value.

**Table 2 antioxidants-12-00450-t002:** Primer sequences used for real-time PCR.

Gene	Forward (5′-3′)	Reverse (5′-3′)	Size (bp)	Accession Number
β-actin	GAGAAATTGTGCGTGACATCA	CCTGAACCTCTCATTGCCA	152	NM_205518.2
Nrf2	CTGCTAGTGGATGGCGAGAC	CTCCGAGTTCTCCCCGAAAG	132	NM_001030756.1
Keap1	GGTTACGATGGGACGGATCA	CACGTAGATCTTGCCCTGGT	135	XM_025145847.1
HO-1	AGCTTCGCACAAGGAGTGTT	GGAGAGGTGGTCAGCATGTC	106	NM_205344.1
NQO1	TCGCCGAGCAGAAGAAGATTGAAG	GGTGGTGAGTGACAGCATGGC	191	NM_001277619.1
GPX1	AGTACATCATCTGGTCGCCG	CTCGATGTCGTCCTGCAGTT	137	NM_001277853.1
Selenof	GAGAACCTTGACTGCTAAC	CACACCTGACATCTGACTA	208	NM_001012926.3
GPX4	ATCCTTACGTGATCGAGAAG	GTGGACAGCAGATACTACAT	249	NM_204220.3
Bax	ATCAATGCAGAGGACCAGGTG	CGTACCGCTTGTTGATGTCG	208	NM_001030920.1
Bcl-2	GATGACCGAGTACCTGAACC	CAGGAGAAATCGAACAAAGGC	114	NM_205339.2
Caspase 3	GAGTTGGAGATGTCCGTTAT	AGTAGGTCTTGTAAGTGATGG	159	XM_046915476.1

Nrf2, nuclear factor erythroid 2-related factor 2; Keap1, Kelch-like ECH-associated protein 1; HO-1, heme oxygenase 1; NQO1, NAD(P)H quinone dehydrogenase 1; GPX1, glutathione peroxidase 1; Selenof, selenoprotein f; GPX4, glutathione peroxidase 4; Bax, BCL2-associated X; Bcl-2, B-cell lymphoma-2.

**Table 3 antioxidants-12-00450-t003:** Effects of different Se sources on laying performance in laying hens ^1^.

Item	Treatments (T) ^2^					SEM	*p*-Value		
	CON	SS	SEY	SEC	SEC + SEY		T	Time	T × Time ^2^
Laying rate (%)									
1–28d	88.60	90.88	90.94	93.14	89.74	0.604	0.074	0.474	0.105
28–56d	89.60	90.10	87.18	93.85	90.16	0.734			
1–56d	89.09	90.46	89.03	93.68	90.03	0.568			
Average egg weight (g)									
1–28d	64.93	64.56	65.33	65.33	65.53	0.220	0.709	0.004	0.538
28–56d	65.53	65.07	65.53	65.97	65.89	0.227			
1–56d	65.22	64.79	65.32	65.62	65.70	0.214			
Average daily egg mass (g/bird per day)									
1–28d	59.53	58.74	59.74	61.34	59.30	0.418	0.303	0.023	0.149
28–56d	59.43	58.60	57.01	61.16	57.79	0.560			
1–56d	59.47	58.69	58.41	61.28	58.49	0.465			
ADFI (g/day/bird)									
1–28d	132.27	129.71	137.82	130.82	126.51	2.641	0.402	0.020	0.777
28–56d	123.29	117.58	129.21	127.73	122.02	2.120			
1–56d	127.36	122.62	133.76	128.91	124.39	1.700			
FCR (g of feed/g of egg)									
1–28d	2.32 ^a^	2.18 ^ab^	2.19 ^ab^	2.02 ^b^	2.03 ^ab^	0.036	0.021	0.482	0.746
28–56d	2.28	2.03	2.19	2.04	2.03	0.040			
1–56d	2.26 ^a^	2.07 ^b^	2.19 ^ab^	2.03 ^b^	2.03 ^b^	0.028			

^1^ Data are means of 6 replicates per dietary treatment. SEM, standard error of mean; ADFI, average daily feed intake; FCR, feed conversion ratio. ^2^ T means treatment and T × Time means the interaction between treatment and time. ^a,b^ Labeled means in a row without a common letter differ, *p* < 0.05.

**Table 4 antioxidants-12-00450-t004:** Effects of different Se sources on egg quality ^1^.

Item	Treatment (T) ^2^					SEM	*p*-Value		
	CON	SS	SEY	SEC	SEC + SEY		T	Time	T × Time ^2^
Egg shape index									
28d	1.32	1.34	1.33	1.33	1.32	0.004	0.336	0.128	0.640
56d	1.35	1.35	1.33	1.34	1.34	0.005			
Eggshell strength, N									
28d	38.13	41.13	40.63	43.29	40.51	0.722	0.045	0.151	0.169
56d	40.39 ^ab^	44.66 ^a^	39.67 ^b^	43.01 ^a^	41.02 ^ab^	0.640			
Albumen height, mm									
28d	6.11	6.18	6.31	6.94	6.58	0.110	0.321	0.439	0.723
56d	6.11	6.26	6.19	6.21	6.20	0.076			
Egg yolk color									
28d	6.57	5.96	6.59	6.30	6.61	0.098	0.152	0.332	0.286
56d	6.00	6.48	6.29	5.74	6.54	0.120			
Egg yolk index									
28d	0.35	0.35	0.35	0.35	0.38	0.003	0.801	0.001	0.266
56d	0.38	0.39	0.38	0.39	0.38	0.004			
Yolk percentage									
28d	26.15	26.09	26.14	27.23	26.25	0.171	0.660	0.001	0.127
56d	28.58	28.17	28.10	27.70	28.38	0.190			
Haugh unit									
28d	70.27 ^b^	77.92 ^a^	75.05 ^ab^	79.78 ^a^	77.92 ^a^	1.050	0.017	0.819	0.161
56d	75.07	76.23	74.39	76.56	76.55	0.567			

^1^ Data are means of 6 replicates of 4 samples each replicate. SEM, standard error of mean. ^2^ T means treatment and T × Time means the interaction between treatment and time. ^a,b^ Labeled means in a row without a common letter differ, *p* < 0.05.

**Table 5 antioxidants-12-00450-t005:** Effects of different Se sources on reproductive organ development in laying hens ^1^.

Item	CON	SS	SEY	SEC	SEC + SEY	SEM	*p*-Value
POFs (n)	5.75	6.25	6.00	6.29	6.13	0.132	0.724
SYFs (n)	10.13	10.25	11.71	12.14	11.25	0.691	0.874
LWFs (n)	35.50 ^b^	41.73 ^ab^	38.00 ^ab^	49.20 ^a^	43.00 ^ab^	1.545	0.048
Ovary index (%)	2.18	2.19	2.24	2.29	2.05	0.041	0.443

^1^ Data are means of 6 replicates per dietary treatment. SEM, standard error of mean; POFs, preovulatory follicles; SYFs, small yellow follicles; LWFs, large white follicles. ^a,b^ Labeled means in a row without a common letter differ, *p* < 0.05.

**Table 6 antioxidants-12-00450-t006:** Effects of different Se sources on ovarian antioxidant capacity in laying hens ^1^.

Item	CON	SS	SEY	SEC	SEC + SEY	SEM	*p*-Value
T-AOC (mmol/g prot)	0.12 ^b^	0.22 ^a^	0.22 ^a^	0.22 ^a^	0.24 ^a^	0.010	<0.001
GSH-PX (U/mg prot)	675.46	812.03	742.60	719.05	884.68	26.726	0.094
MDA (nmol/mg prot)	0.71 ^a^	0.66 ^ab^	0.52 ^b^	0.50 ^b^	0.60 ^ab^	0.026	0.046
T-SOD (U/mg prot)	764.46	800.35	836.41	798.87	891.86	23.070	0.469

^1^ Data are means of 6 replicates per dietary treatment. SEM, standard error of mean; T-AOC, total antioxidant capacity; GSH-PX, glutathione peroxidase; MDA, malondialdehyde; T-SOD, total superoxide dismutase. ^a,b^ Labeled means in a row without a common letter differ, *p* < 0.05, whereas *p* < 0.10 was used as the criterion for tendency.

## Data Availability

Data are contained within the article.

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
