# Peer review of "Se-Enriched Cardamine violifolia Improves Laying Performance and Regulates Ovarian Antioxidative Function in Aging Laying Hens"

_antioxidants, 2023, doi:10.3390/antiox12020450_

Round 1

Reviewer 1 Report

Cardamine violifolia (SEC) is a selenium-rich plant with excellent antioxidant function. This manuscript studied the effects of SEC on laying performance and ovarian antioxidant capacity of aging laying hens. Overall, the manuscript is well written and the experiment design is precise. The topic chosen for this study was novelty, meaningful for the poultry breeding and anti-aging. However, I have the following comments and questions on the manuscript which the authors should consider.

Abstract

1. Every reader know selenium (Se) is an essential trace element. The author can add the characteristics of SEC, such as “The main form of SEC is organic Se, which mainly exists in the form of Se-enriched protein. The edibility and Se accumulation ability of SEC is expected to develop new sources of organic Se supplementation for human and animal nutrition.” The research background needs to be supplemented.

2. “Results showed that SEC improved laying performance and egg quality.” Specific significance results should be shown here and should be repeated by the authors at the end of the abstract.

Introduction

3. “Based on this, the effective measure to alleviate ovarian aging is to reduce oxidative stress.” This is not validated, and the above description in front of this sentence can only conclude that oxidative stress is a major factor in ovarian aging, and there is no literature proving that effective measures are inhibiting oxidative stress. Ovarian senescence should be based on the quantity and quality of follicular atresia and oocytes, and these related literatures need to be supplemented.

Sample collection

4. The author had 5 treatments with 6 replicates per treatment. How were 8 chickens selected for sampling?

Tables and figures

5. “Follicles number is shown in Table 5.” There is no table 5 in this manuscript, author missed this table.

6. There are some mistakes in table 3,4 and 6. For average daily egg mess of 28-56d, it should not label superscript letters for significant differences, because the P value > 0.05. Same mistakes in table 4 and 6.  

Discussion

7. “Therefore, the effective measures for retarding ovarian aging is to reduce oxidative stress.” This conclusion is not acceptable because there is a lack of literature to prove that reducing oxidative stress delays ovarian aging.

Conclusion

8. Ovarian senescence is the main reason for the decrease of laying rate and egg quality. The author should reflect the influence of SEC on ovarian senescence in the conclusion. Such as “In conclusion, dietary Se-enriched Cardamine violifolia exerted antioxidant function through the modulation of the Nrf2/Keap1 signaling pathway, alleviates ovarian senescence, and improves laying rate and egg quality. Furthermore, this study provides evidence for the development and application of Cardamine violifolia as feed additives for improving laying performance and egg quality in laying hens.”

References

9. References 18 and 21, 28, 36, 42 needs to be modified. The author needs to check the citation to make sure it meets the journal's requirements, so it will be accepted more quickly.

Author Response

1) Every reader knows selenium (Se) is an essential trace element. The author can add the characteristics of SEC, such as “The main from of SEC is organic Se, which mainly exists in the form of Se-enriched protein. The edibility and Se accumulation ability of SEC is expected to develop new sources of organic Se supplementation for human and animal nutrition.” The research background needs to be supplemented.

Response: Thank you for your suggestion. Due to word limit (The abstract should be total of about 200 words maximum), we added “The edibility of SEC is expected to develop new sources of organic Se supplementation for human and animal nutrition” and deleted “Selenium (Se) is an essential trace element for human and animal health.”

2) “Result showed that SEC improved laying performance and egg quality.” Specific significance results should be shown here and should be repeated by the authors at the end of the abstract.

Response: Thank you for your suggestion. We changed “Result showed that SEC improved laying performance and egg quality” to “Result showed that SEC improved rate and Haugh unit of eggs, and reduced FCR.” And we repeated the results at the end of the abstract.

3) “Based on this, the effective measure to alleviate ovarian aging is to reduce oxidative stress.” This is not validated, and the above description in front of this sentence can only conclude that oxidative stress is a major factor in ovarian aging, and there is no literature proving that effective measures are inhibiting oxidative stress. Ovarian senescence should be based on the quantity and quality of follicular atresia and oocytes, and these related literatures need to be supplemented.

Response: Thank you for your suggestion. In fact, among all the inducing factors of ovarian aging, oxidative stress is one of the most dominant factors [1-3]. Therefore, we revised this sentence “Based on this, it has been explored in poultry production to alleviate ovarian aging by reducing oxidative stress.” Ovarian aging is characterized by follicular atresia and decrease in both the quantity and the quality of oocytes [4]. We added related literatures. The relevant content is in page 1 line 39-40 and page 2 line 53-54.

4) The author had 5 treatments with 6 replicates per treatment. How were 8 chickens selected for sampling?

Response: Thank you very much for your comments. We are very sorry that it was our negligence. At the end of experiment, 6 hens per treatment (one hen each replicate) were weight and slaughtered. We have revised in page 3 line 106.

5) “Follicles number is shown in Table 5.” There is no table 5 in this manuscript, author missed this table.

Response: Thank you very much for your comments. We are very sorry that we missed this table. We added it in page 7.

6) There are some mistakes in table 3, 4 and 6. For average daily egg mass of 28-56d, it should not label superscript letters for significant differences, because the P value > 0.05. Same mistakes in table 4 and 6.

Response: Thank you very much for your comments. We checked carefully data. Data for laying performance and egg quality traits were analyzed using repeated measure analysis in the general linear model to evaluate the effects of time and treatments. P < 0.05 was defined as significance while p < 0.10 was considered to be a trend toward significance. We revised the table 3, 4, 6 and related results.

7) “Therefore, the effective measures for retarding ovarian aging is to reduce oxidative stress.” This conclusion is not acceptable because there is a lack of literature to prove that reducing oxidative stress delays ovarian aging.

Response: Thank you for your suggestion. We agree with you. Although numerous studies have documented that oxidative stress is a leading driver of the ovarian aging process [5-7], there is a lack of literature to prove that reducing oxidative stress delays ovarian aging. Therefore, we changed this sentence to “Therefore, alleviating oxidative stress in the ovaries might be an important breakthrough for retarding ovarian aging.”

8) Ovarian senescence is the main reason for the decrease of laying rate and egg quality. The author should reflect the influence of SEC on ovarian senescence in the conclusion. Such as “In conclusion, dietary Se-enriched Cardemine violifolia exerted antioxidant function through the modulation of the Nrf2/Keap1 signaling pathway, alleviates ovarian senescence, and improves laying rate and egg quality. Furthermore, this study provides evidence for the development and application of Cardamine violifolia as feed additives for improving laying performance and egg quality in laying hens.

Response: Thank you for your suggestion. We revised the conclusion. The relevant content is in page 12 line 332-337.

9) Reference 18 and 21, 28, 36, 42 needs to be modified. The author needs to check the citation to make sure it meets the journal’s requirements, so it will be accepted more quickly.

Response: Thank you for your suggestion. We checked the citation and modified reference 18 and 21, 28, 36, 42.

References

  1. Devine, P.J.; Perreault, S.D.; Luderer, U. Roles of reactive oxygen species and antioxidants in ovarian toxicity. Biol Reprod. 2012, 86,
  2. Luderer, U. Ovarian toxicity from reactive oxygen species. Vitam Horm. 2014, 94, 99–127.
  3. Tesarik, J.; Galán-Lázaro, M.; Mendoza-Tesarik, R. Ovarian aging: Molecular mechanisms and medical management. Int J Mol Sci. 2021, 22, 1371
  4. Yao, H.H.; Volentine, K.K.; Bahr, J.M. Destruction of the germinal disc region of an immature preovulatory chicken follicle induces atresia and apoptosis. Biol Reprod. 1998, 59, 516–521.
  5. Yang, H.X.; Qazi, I.H.; Pan, B.; Angel, C.; Guo, S.C.; Yang, J.Y.; Zhang, Y.; Ming, Z.; Zeng, C.J.; Meng, Q.Y.; et al. Dietary selenium supplementation ameliorates female reproductive efficiency in aging mice. Antioxidants 2019, 8, 634.
  6. Agarwal, A.; Gupta, S.; Sharma, R.K. Role of oxidative stress in female reproduction. Biol Endocrinol. 2005, 3, 28.
  7. Lim, J.; Luderer, U. Oxidative damage increases and antioxidant gene expression decreases with aging in the mouse ovary. Biol Reprod. 2011, 84, 775–782.

Reviewer 2 Report

I have indicated reconsider after revision and re-submission. The basis of my suggestion is based on the concern I have with the statistical analysis of the data. A one-way ANOVA test for differences of means is used as the  fundamental analysis. For laying performance measures for most of the variables the P values indicate non-significant effects between means, but the authors continue to run a post hoc Duncan’s multiple comparison test and then identify differences between individual means. If the P value of the ANOVA is greater then the 0.05 then the null hypothesis needs to be accepted i.e. The null hypothesis states that the population means are all equal. Based on the P value <0.05 it is not appropriate to further analyse for individual effects.

For example

These are the treatment values for egg production over days 1-56 of the experiment

1-56d  89.09b   90.46ab   89.03b   93.68a  90.03ab   0.641     P = 0.125

The P value indicates that the null hypothesis is accepted and that there are no differences between the mean values even though there appear to be numerical differences.

The same issue exists for the egg quality measures.

For sections

3.3. Ovarian follicle development

3.4. Proliferation and apoptosis in ovary

3.4. Proliferation and apoptosis in ovary

3.4. mRNA expression of selenoproteins and Nrf2/HO-1 pathway related genes in ovary of hens

The authors provided the individual differences following the Ad hoc testing but fail to provide the P value for the ANOVA. It is not possible to evaluate the relevance of the individual comparisons without knowing the outcome of the ANOVA.

Maybe the authors need to consider a different statistical analysis. Having repeated measures maybe use REML with treatment and week as the main effects.

I have included some grammatical considerations of the introduction. Methods and Results. I have not evaluated the discussion based on my concern for the statistical analysis.

The work is extensive and warrants further consideration after a revised submission

Author Response

I have indicated reconsider after revision and re-submission. The basis of my suggestion is based on the concern I have with the statistical analysis of the data. A one-way ANOVA test for differences of means is used as the fundamental analysis. For laying performance measures for most of the variables the P values indicate non-significant effects between means, but the authors continue to run a post hoc Duncan’s multiple comparison test and then identify differences between individual means. If the P value of the ANOVA is greater then the 0.05 then the null hypothesis needs to be accepted i.e. The null hypothesis states that the population means are all equal. Based on the P value <0.05 it is not appropriate to further analyse for individual effects.

For example

These are the treatment values for egg production over days 1-56 of the experiment

1-56d  89.09b   90.46ab   89.03b   93.68a  90.03ab   0.641     P = 0.125

The P value indicates that the null hypothesis is accepted and that there are no differences between the mean values even though there appear to be numerical differences.

The same issue exists for the egg quality measures.

For sections

3.3. Ovarian follicle development

3.4. Proliferation and apoptosis in ovary

3.4. Proliferation and apoptosis in ovary

3.4. mRNA expression of selenoproteins and Nrf2/HO-1 pathway related genes in ovary of hens

The authors provided the individual differences following the Ad hoc testing but fail to provide the P value for the ANOVA. It is not possible to evaluate the relevance of the individual comparisons without knowing the outcome of the ANOVA.

Maybe the authors need to consider a different statistical analysis. Having repeated measures maybe use REML with treatment and week as the main effects.

I have included some grammatical considerations of the introduction. Methods and Results. I have not evaluated the discussion based on my concern for the statistical analysis.

The work is extensive and warrants further consideration after a revised submission.

Response: Thank you for your suggestion. We check all data. The data were analyzed again. The experimental data including ovarian follicle development, follicular atretic rate, ovarian cell proliferation and apoptosis, ovarian antioxidant parameters and mRNA expression of selenoproteins and Nrf2/Keap1 were evaluated using one-way analysis of variance (ANOVA) followed by Duncan’s multiple comparison test. Data for laying performance and egg quality were analyzed using repeated measure analysis in the general linear model to evaluate the effects of time and different treatments. P < 0.05 was defined as significance while p < 0.10 was considered to be a trend toward significance. According to statistical analysis, we found that the laying rate was increased (p= 0.048) in hens fed SEC supplemented diet compared with the CON group during d 1-56 and it had a trend to increase (p = 0.062) during d 28-56. We revised the table 3, 4, 6 and labeled superscript letters for significant differences. According to statistical analysis, we revised relevant results.

Because some data were shown as column graphs, P values were not provided. Therefore, we supplied tables of all graphs. Furthermore, we revised grammatical errors in the Introduction. Methods, Results and Discussion.

Table 1. Effects of different Se sources on follicular atretic rate and ovarian proliferation and apoptosis in in laying hens1.

Items

CON

SS

SEY

SEC

SEC+SEY

SEM

p-Value

Follicular atretic rate (%)

17.89a

15.00ab

11.82ab

7.57ab

6.51b

1.355

0.047

PCNA index

5.40c

9.11b

9.73ab

9.85ab

12.46a

0.519

0.002

TUNEL index

3.74a

2.93ab

2.51bc

1.66c

1.78c

0.154

< 0.001

Bax

1.00ab

1.26a

1.02ab

0.79bc

0.53c

0.056

< 0.001

Caspse3

1.00ab

1.12a

0.81bc

0.68c

0.83bc

0.043

0.005

Bcl1

1.00c

1.46ab

1.30b

1.63a

1.32b

0.052

0.001

1Data are means of 6 replicates per dietary treatment. SEM, standard error of mean. Bax, BCL2-associated X; Bcl-2, B-cell lymphoma-2.

abLabeled means in a row without a common letter differ, p < 0.05.

Table 2. Relative mRNA expression of selenoprotein and Nrf2/Keap1 pathway genes in ovary of laying hens fed different Se sources1.

Items

CON

SS

SEY

SEC

SEC+SEY

SEM

p-Value

Selenof

1.00c

1.32b

1.61b

1.99a

1.41b

0.070

< 0.001

GPX1

1.00b

1.62a

1.87a

1.49a

1.80a

0.082

0.003

GPX4

1.00b

0.94b

1.16b

1.46a

1.15b

0.050

0.006

Nrf2

1.00c

1.55abc

1.37bc

1.93ab

1.97a

0.100

0.005

Keap1

1.00a

0.92ab

0.77ab

0.65b

0.78ab

0.040

0.048

NQO1

1.00c

1.25bc

1.92a

1.58ab

1.30bc

0.076

< 0.001

HO-1

1.00c

1.12bc

1.42ab

1.57a

1.47ab

0.060

0.006

1Data are means of 6 replicates per dietary treatment. SEM, standard error of mean. Selenof, selenoprotein f; GPX1, glutathione peroxidase 1; GPX4, glutathione peroxidase 4; Nrf2, nuclear factor erythroid 2 like 2; Keap1, kelch like ECH-associated protein 1; NQO1, NAD(P)H quinone dehydrogenase 1; HO-1, heme oxygenase 1.

abcLabeled means in a row without a common letter differ, p < 0.05.

Round 2

Reviewer 1 Report

I have no more comments for this manuscript. 

Author Response

I have no more comments for this manuscript.

Response: Thank you very much for your reviewing work. We are pleased to hearing your satisfaction for our revision.

We hope we have adequately dealt with all of the concerns and look forward to hearing from you in the near furture. If needed, we would be pleased to answer any further questions.

Reviewer 2 Report

There remains a concern with the authors interpretation of the statistical analysis of the egg laying performance. This is important as the results of the laying performance are strongly linked by the authors to the other measures of ovarian aging (follicular cell health and oxidative status). In the revised submission the authors have used a REML analysis to determine the level of significance of the laying rate results. The probability value is 0.074 which suggest a strong trend but not significance. For some reason the authors have accepted as 0.074 as sufficient evidence to run a Duncan test and identify significant effects between individual treatments and used this as support for a link between the measures of follicular cell health and oxidative status and increased laying rate in SEC treated hens.

This is stated at:

Ln 24: ‘Results showed that SEC improved laying rate and Haugh unit of eggs, and reduced FCR’

Ln 28: ‘These results indicate that SEC improves laying performance and egg quality by enhancement of ovarian antioxidant capacity.

Ln 275; ‘However, dietary SEC supplementation increased laying rate and decreased FCR of laying hens. Similar results have not been reported so far

It would  be more appropriate to state

Ln 24: ‘Supplementation with SEC tended to increase laying rate and increased Haugh unit of eggs, and reduced FCR’

Other comments are

Ln 24: ‘Results showed that SEC improved laying rate and Haugh unit of eggs, and reduced FCR’

Ln 28: ‘These results indicate that SEC improves laying performance and egg quality by enhancement of ovarian antioxidant capacity.

This needs to be considered in-light of later comments about the statistical analysis.

Ln 47: remove ‘system is one of the most” add ‘systems are’

Ln 48: remove ‘an inactive form’ add ‘as an inactive form’

Ln 49: remove ‘activates expressions’ add ‘activating expression’

Ln 51: remove ‘Based on this, it has been explored in poultry production to alleviate ovarian aging by reducing oxidative stress’

add ‘Based on this, alleviating ovarian aging by reducing oxidative stress has been research in poultry’

Ln 60: remove ‘diet’ add ‘diet in mice’

Ln 66: remove ‘retard ovarian aging’ add ‘acting to retard ovarian aging’

Ln 96: remove ‘freshly eggs’ add ‘freshly laid eggs’

Ln 104: remove ´one hen each replicate’ add ‘one hen from each replicate’

Ln 108: ‘1mm’ add space ‘1 mm’

Ln 113: remove ‘section’ add ‘sections’

Ln 22: remove ‘ add ‘in the ovary’

Ln 143 & 147: remove ‘labeling’ add ‘labelled’

Ln 154: here the authors use P and p for the probabilities. Need to be consistent with journal acceptance as p is used in the results

Ln 158: ‘The laying performance of hens is presented in Table 3. The laying rate was increased (p < 0.05) in hens fed SEC supplemented diet compared with the CON group during d 1-56 and it had a trend to increase (p < 0.10) during d 28-56. Moreover, hens fed SEC diet had higher (p < 0.05) laying rate than those fed SEY and SEC + SEY diet during d 1-56’

This is confusing. In the table the P given is for T is 0.074 and the T X Time is 1.05. There is a non-significant treatment effect (0.74) and there is no change over time as the P for T x Time is 0.105.

Why have the authors found individual differences at day 28-56 and 1-56? Based on no time effect

The laying rate values seem different.

Have the authors check for normality of the data and could a transformation of the data (log, Ln, etc) improve the analysis?

Important comment:

As with the GSH-PX (U/mg prot) are the authors accepting P < 0.01) as a trend and then progressing to look at the individual effects? The questions to ask is this accepted by the journal as normal practice.

Ln 137: ‘Dietary SEC and SEC + SEY supplementation increased (p < 0.05) Haugh unit compared with 174 hens fed CON diet at d 28 (Table 4).’ NOTE - Also SS is  higher than Con

Also why is the difference only at d 28 as there is a non significant T x time interaction which would indicate that the difference between treatments is not time dependent

Ln 261: remove ‘oocytes’ add ‘oocyte’

Ln 271: remove ‘The edibility and Se accumulation ability of SEC is expected to develop’ add ‘The edibleness and readily accumulation of Se from SEC is expected to support further development of’

Ln 275; ‘However, dietary SEC supplementation increased laying rate and decreased FCR of laying hens. Similar results have not been reported so far. Furthermore, SEC or SEC + SEY supplementation increased Haugh unit. These results suggested that SEC could improve laying performance and egg quality in the late phase.’

This needs to be considered in relation to the comments made about the statistical analysis discussed above. There are concerns with these conclusions

Ln 281: remove ‘follicles.’ Add ‘ovarian follicles.’

Ln 286: remove ‘factor’ add ‘factors’

Ln 293: remove ‘and improved’ add ‘could potentially improve’

 Maybe that this is more aligned with the trend for laying performance to increase based on the P value of 0.074

I suggest this based on the possible concerns I have with the statistical analysis of laying rate

Ln 302: remove ‘SEC and SEY or SEC + SEY’ add ‘SEC, SEY and SEC + SEY’

Ln 309: remove ‘during the follicle development’ add ‘during ovarian follicle development’

Ln 313 & 314: remove ‘ameliorated’ add ‘ameliorating’

Author Response

Dear reviewer,

Thank you very much for your comments. I am very grateful to your valuable suggestions on my submitted manuscript ID: antioxidants-2099649, entitled "Se-enrich Cardamine violifolia improves laying performance and regulates ovarian antioxidative function in aging laying hen”. I believe they are very helpful and important. Based on your comments and requests, we have made extensive modification on the manuscript. The revision has been marked in red in the revised manuscript. All the questions were answered as follows.

List of actions:

Comment

1) There remains a concern with the authors interpretation of the statistical analysis of the egg laying performance. This is important as the results of the laying performance are strongly linked by the authors to the other measures of ovarian aging (follicular cell health and oxidative status). In the revised submission the authors have used a REML analysis to determine the level of significance of the laying rate results. The probability value is 0.074 which suggest a strong trend but not significance. For some reason the authors have accepted as 0.074 as sufficient evidence to run a Duncan test and identify significant effects between individual treatments and used this as support for a link between the measures of follicular cell health and oxidative status and increased laying rate in SEC treated hens.

Response: Thank you for your suggestion. We believe that your suggestions are very valuable. Laying performance are important. The laying performance includes laying rate and FCR. In the results, laying rate was tended to increase (P = 0.074), but the FCR reduced (P < 0.05). We combined the results of laying rate and FCR in the conclusion. We are very sorry that it was our negligence. We revised the relevant results in page 1 line 24 and line 29.

2) This is stated at:

Ln 24: ‘Results showed that SEC improved laying rate and Haugh unit of eggs, and reduced FCR’

Ln 28: ‘These results indicate that SEC improves laying performance and egg quality by enhancement of ovarian antioxidant capacity.

Ln 275; ‘However, dietary SEC supplementation increased laying rate and decreased FCR of laying hens. Similar results have not been reported so far

It would be more appropriate to state

Ln 24: ‘Supplementation with SEC tended to increase laying rate and increased Haugh unit of eggs, and reduced FCR’

Other comments are

Ln 24: ‘Results showed that SEC improved laying rate and Haugh unit of eggs, and reduced FCR’

Ln 28: ‘These results indicate that SEC improves laying performance and egg quality by enhancement of ovarian antioxidant capacity.’

This needs to be considered in-light of later comments about the statistical analysis.

Response: Thank you for your suggestion. According to the later comments about the statistical analysis, we revised relevant content in the line 24, 29 and 275-279.

3) Ln 47: remove ‘system is one of the most” add ‘systems are’

Response: Thank you for your suggestion. We removed “system is one of the most” and added “systems are”.

4) Ln 48: remove ‘an inactive form’ add ‘as an inactive form’

Response: Thank you for your comment. We removed “an inactive form” and added “as an inactive form”.

5) Ln 49: remove ‘activates expressions’ add ‘activating expression’

Response: Thank you for your comment. We removed “activates expressions” and added “activating expression”.

6) Ln 51: remove ‘Based on this, it has been explored in poultry production to alleviate ovarian aging by reducing oxidative stress’ add ‘Based on this, alleviating ovarian aging by reducing oxidative stress has been research in poultry’

Response: Thank you for your comment. We removed “Based on this, it has been explored in poultry production to alleviate ovarian aging by reducing oxidative stress” and added “Based on this, alleviating ovarian aging by reducing oxidative stress has been researched in poultry”.

7) Ln 60: remove ‘diet’ add ‘diet in mice’

Response: Thank you for your comment. We removed “diet” and added “diet in mice”.

8) Ln 66: remove ‘retard ovarian aging’ add ‘acting to retard ovarian aging’

Response: Thank you for your comment. We removed “retard ovarian aging” and added “acting to retard ovarian aging”.

9) Ln 96: remove ‘freshly eggs’ add ‘freshly laid eggs’

Response: Thank you for your comment. We removed “freshly eggs” and added “freshly laid eggs”.

10) Ln 104: remove ´one hen each replicate’ add ‘one hen from each replicate’

Response: Thank you for your comment. We removed “one hen each replicate” and added “one hen from each replicate.

11) Ln 108: ‘1mm’ add space ‘1 mm’

Response: Thank you for your suggestion. We added space between “1” and “mm”.

12) Ln 113: remove ‘section’ add ‘sections’

Response: Thank you for your suggestion. We removed “section” and added “sections”.

13) Ln 122: remove ‘add’ in the ovary’

Response: Thank you for your suggestion. We changed “in ovary” to “in the ovary”.

14) Ln 143 & 147: remove ‘labeling’ add ‘labelled’

Response: Thank you for your suggestion. We changed “labeling” to “labelled” in line 145 and 149.

15) Ln 154: here the authors use P and p for the probabilities. Need to be consistent with journal acceptance as p is used in the results.

Response: Thank you for your suggestion. We changed “P” to “p”, and checked the results.

16) Ln 158: ‘The laying performance of hens is presented in Table 3. The laying rate was increased (p < 0.05) in hens fed SEC supplemented diet compared with the CON group during d 1-56 and it had a trend to increase (p < 0.10) during d 28-56. Moreover, hens fed SEC diet had higher (p < 0.05) laying rate than those fed SEY and SEC + SEY diet during d 1-56’

This is confusing. In the table the P given is for T is 0.074 and the T X Time is 1.05. There is a non-significant treatment effect (0.74) and there is no change over time as the P for T x Time is 0.105.

Why have the authors found individual differences at day 28-56 and 1-56? Based on no time effect. The laying rate values seem different.

Have the authors check for normality of the data and could a transformation of the data (log, Ln, etc) improve the analysis?

Response: Thank you for your comment. We believe that your suggestions are very valuable. We have checked the data and calculated the laying rate again. In the table the P given is for T is 0.074 and the T X Time is 1.05. There is a non-significant treatment effect (0.74) and there is no change over time as the P for T x Time is 0.105. We are very sorry that it was our negligence. The laying rate tended to increase (p < 0.1) in hens fed SEC supplemented diet compared with the CON group. Moreover, hens fed SEC diet had a trend to increase (p < 0.1) laying rate compared with those fed SEY and SEC + SEY diet. We revised the relevant the results.

17) Important comment:

As with the GSH-PX (U/mg prot) are the authors accepting P < 0.10) as a trend and then progressing to look at the individual effects? The questions to ask is this accepted by the journal as normal practice.

Response: Thank you for your comment. We believe that your suggestions are very valuable. We deleted superscripts letter of GSH-PX, and added “whereas p < 0.10 was used as the criteria for tendency” in the note.

18) Ln 173: ‘Dietary SEC and SEC + SEY supplementation increased (p < 0.05) Haugh unit compared with hens fed CON diet at d 28 (Table 4).’ NOTE - Also SS is higher than Con Also why is the difference only at d 28 as there is a no significant T x time interaction which would indicate that the difference between treatments is not time dependent.

Response: Thank you for your comment. We changed to “Dietary SEC, SEC + SEY and SS supplementation increased (p < 0.05) Haugh unit compared with hens fed CON diet”. The relevant content is in page 6 line 173-174.

19) Ln 261: remove ‘oocytes’ add ‘oocyte’

Response: Thank you for your suggestion. We removed “oocytes” and added “oocyte.”

20) Ln 271: remove ‘The edibility and Se accumulation ability of SEC is expected to develop’ add ‘The edibleness and readily accumulation of Se from SEC is expected to support further development of’

Response: Thank you for your comment. We removed “The edibility and Se accumulation ability of SEC is expected to develop” and added “The edibleness and readily accumulation of Se from SEC is expected to support further development of new organic Se supplementation for human and animal nutrition”.

21) Ln 275; ‘However, dietary SEC supplementation increased laying rate and decreased FCR of laying hens. Similar results have not been reported so far. Furthermore, SEC or SEC + SEY supplementation increased Haugh unit. These results suggested that SEC could improve laying performance and egg quality in the late phase.’

This needs to be considered in relation to the comments made about the statistical analysis discussed above. There are concerns with these conclusions.

Response: Thank you for your comment. According to the later comments about the statistical analysis, we changed to “However, dietary SEC supplementation tended to increase laying rate and decreased FCR of laying hens. Similar results have not been reported so far. Furthermore, SEC or SEC + SEY supplementation increased Haugh unit. These results suggested that SEC could potentially improve laying performance and egg quality in the late phase”.

22) Ln 281: remove follicles.’ Add ‘ovarian follicles.’

Response: Thank you for your comment. We removed “follicles” and added to “ovarian follicles”.

23) Ln 286: remove ‘factor’ add ‘factors’

Response: Thank you for your comment. We removed “factor” and added “factors.”

24) Ln 293: remove ‘and improved’ add ‘could potentially improve’ Maybe that this is more aligned with the trend for laying performance to increase based on the P value of 0.074.

Response: Thank you for your comment. We removed “and improved” and added “could potentially improve”.

25) Ln 302: remove ‘SEC and SEY or SEC + SEY’ add ‘SEC, SEY and SEC + SEY’

Response: Thank you for your comment. We removed “SEC and SEY or SEC + SEY” and added “SEC, SEY and SEC + SEY”.

26) Ln 309: remove ‘during the follicle development’ add ‘during ovarian follicle development’

Response: Thank you for your comment. We removed “during the follicle development” and added “during ovarian follicle development”.

27) Ln 313 & 314: remove ‘ameliorated’ add ‘ameliorating’

Response: Thank you for your suggestion. We removed “ameliorated and alleviated” and added “ameliorating and alleviating”.

Round 3

Reviewer 2 Report

The paper is comprehensive and detailed. I reads well. I have just one comment with the statistical analysis following on from my first review.

Results

Table 3. In my previous review I indicated that authors needed to remove the individual comparisons for egg laying as the p value was 0.072 and so the individual comparisons were not relevant. However, the authors have seen this comment pertaining to all analyses. The individual comparisons for FCR in table 3 and for and HU in Table 4. The authors should reinstall the superscripts identifying the individual differences in these cases.

Author Response

The paper is comprehensive and detailed. I read well. I have just one comment with the statistical analysis following on from my first review.

Results

Table 3. In my previous review I indicated that authors needed to remove the individual comparisons for egg laying as the p value was 0.072 and so the individual comparisons were not relevant. However, the authors have seen this comment pertaining to all analyses. The individual comparisons for FCR in table 3 and for HU in Table 4. The authors should reinstall the superscripts identifying the individual differences in these cases.

Response: Thank you very much for your reviewing work.

We have removed the individual comparisons for egg laying rate. According to the statistical analysis, we reinstall the superscripts for FCR in table 3 and for HU in table 4. We added “abLabeled means in a row without a common letter differ, p < 0.05” in the note of table 3 and table 4.

We hope we have adequately dealt with all of the concerns and look forward to hearing from you in the near future. If needed, we would be pleased to answer any further questions.